# On-Chip Broadband, Compact TM Mode Mach–Zehnder Optical Isolator Based on InP-on-Insulator Platforms

**DOI:** 10.3390/nano14080709

**Published:** 2024-04-18

**Authors:** Wan-Ting Chen, Li Liu, Jia Zhao, Chen Zhang

**Affiliations:** 1School of Information Science and Engineering, Shandong University, Qingdao 266237, China; 202234052@mail.sdu.edu.cn (W.-T.C.); liulisddx@mail.sdu.edu.cn (L.L.); 2School of Key Laboratory of Laser & Infrared System, Ministry of Education, Shandong University, Qingdao 266237, China

**Keywords:** optical isolator, non-reciprocal phase shift, magneto-optical, Mach–Zehnder interferometer, InP-on-insulator

## Abstract

An integrated optical isolator is a crucial part of photonic integrated circuits (PICs). Existing optical isolators, predominantly based on the silicon-on-insulator (SOI) platform, face challenges in integrating with active devices. We propose a broadband, compact TM mode Mach–Zehnder optical isolator based on InP-on-insulator platforms. We designed two distinct magneto-optical waveguide structures, employing different methods for bonding Ce:YIG and InP, namely O_2_ plasma surface activation direct wafer bonding and DVS-benzocyclobutene (BCB) adhesive bonding. Detailed calculations and optimizations were conducted to enhance their non-reciprocal phase shift (NRPS). At a wavelength of 1550 nm, the direct-bonded waveguide structure achieved a 30 dB bandwidth of 72 nm with a length difference of 0.256 µm. The effects of waveguide arm length, fabrication accuracy, and dimensional errors on the device performance are discussed. Additionally, manufacturing tolerances for three types of lithographic processes were calculated, serving as references for practical manufacturing purposes.

## 1. Introduction

An optical isolator is a critical non-reciprocal component in optical communication systems [1]. It prevents interference caused by reflected light, thus protecting active components and enhancing system stability [2,3,4]. Optical isolation can be achieved through magneto-optical (MO) effects [5,6,7,8], nonlinear photonic effects [9,10,11], and spatio-temporal modulation [12,13,14]. Among them, MO isolators are highly promising, owing to their simple device structure, low insertion loss, and large isolation bandwidth. MO materials, such as yttrium iron garnet (YIG), exhibit non-reciprocal phase shifts (NRPSs) due to their asymmetric dielectric constant matrix after magnetization [15,16,17,18]. YIG doped with cerium (Ce) or bismuth (Bi) has a large Faraday rotation coefficient and low optical absorption loss in the near-infrared band, making it a good candidate for optical isolation in communication systems [19,20,21].

Currently, integrating MO isolators with CMOS-compatible semiconductor platforms remains challenging due to factors such as lattice mismatch and shape-induced birefringence. The most widely studied MO isolators are based on silicon-on-insulator (SOI) platforms [22,23,24,25,26]. SOI waveguides are compatible with CMOS technology but the integration of light sources made of III-V compound semiconductors with SOI platforms is difficult [27,28,29]. Heterogeneous integration of both MO isolators and III-V compound semiconductor lasers with Si photonics platforms is important but has rarely been studied.

Indium phosphide (InP) is widely used in low-loss optical communication fields at 1310 nm and 1550 nm wavelengths due to its high refractive index and photoelectric conversion efficiency [30,31,32]. Heterogeneous integration of InP-based active-layer lasers with SiO_2_/Si substrates has reached the commercial stage [33,34,35]. Reniers et al. developed an on-chip magneto-optical circulator on the InP-on-insulator platform with good performance [36]. However, there remains an absence of development and simulation of integrated optical isolators.

In this work, we compare two NRPS waveguide structures, employing different methods for bonding Ce:YIG and InP, namely O_2_ plasma surface activation direct wafer bonding and DVS-benzocyclobutene (BCB) adhesive bonding. We demonstrate a fundamental TM mode optical isolator based on the InP-on-insulator platform. The device adopts an asymmetric Mach–Zehnder interferometer (MZI) structure, composed of two 1 × 2 multi-mode interferometric (MMI) couplers, asymmetric reciprocal phase shift (RPS) waveguides, and NRPS waveguides. We design two non-reciprocal waveguide structures based on the direct bonding process and the BCB bonding process. The variation in NRPS with different InP thicknesses and widths of the two waveguide structures is calculated and compared. The maximum NRPS of direct-bonded waveguides is 6488.34 rad/m, which is three times greater than that of BCB-bonded waveguides. This result is attributed to the longitudinal distribution gradient of H_x_. To meet the needs of device miniaturization, we choose the direct-bonded waveguide structure for our subsequent calculations. The impact of varying lengths of RPS waveguides on both isolation and bandwidth is discussed. Noteworthy, the 30 dB isolation bandwidth exceeds 70 nm and the isolation is much greater than 50 dB when the length difference of RPS waveguides is supposedly 0.256 µm. In addition, we discuss the impact of size error with the waveguide arms and the InP waveguide on the isolation performance and the shift in the transmission spectra, which was rarely mentioned in previous studies. Finally, the manufacturing tolerances of three process precisions are simulated, which can be used as references in future experiments.

## 2. Device Structure and Principle

The magnetization of MO material induces the MO effect, which is utilized for phase modulation to achieve isolation. The dielectric constant tensor of an MO material magnetized along the *x* direction can be expressed as follows:(1)$$\varepsilon =\left[\begin{array}{ccc} \varepsilon_{x} & 0 & 0\\ 0 & \varepsilon_{y} & j\gamma \\ 0 & -j\gamma & \varepsilon_{z} \end{array}\right]$$
where $\varepsilon_{x}=\varepsilon_{y}=\varepsilon_{z}={n_{Ce:YIG}}^{2}$, and γ is the off-diagonal term of the dielectric constant tensor, which is determined by the refractive index $n_{Ce:YIG}$ and the Faraday rotation $\theta_{F}$ of the MO material, as $2n_{Ce:YIG}\theta_{F}/k_{0}$, where $k_{0}$ is the wavenumber in vacuum.

Due to the presence of the off-diagonal term in the dielectric constant matrix, light propagating along the *z*-axis in the MO waveguide exhibits varying propagation constants $\left(\beta_{f},\beta_{b}\right)$ based on the direction of propagation or magnetization. The NRPS is defined as the difference between propagation constants ($\Delta \beta_{TM}$). The relationship between the NRPS and the dielectric constant tensor can be derived by incorporating it into Maxwell’s equations and applying the perturbation principle [15,37].
(2)$$NRPS=\Delta \beta_{TM}=\frac{2\beta_{TM}}{\omega \varepsilon_{0}p}\iint \frac{\gamma }{n_{0}^{4}}H_{x}\partial_{y}H_{x}dxdy$$
where $P={\left(\iint E\times H^{*}+E^{*}\times H\right)}_{Z}dxdy$ is the normalized power flow along the z direction, $\beta_{TM}$ is the propagation constant for the fundamental TM mode, and ω and $\varepsilon_{0}$ are the frequency and vacuum dielectric constant, respectively.

The overall structure of the asymmetric MZI isolator is shown in Figure 1a. It consists of RPS waveguides, NRPS waveguides, and two 3 dB 1 × 2 MMI couplers. A permanent magnet is positioned on one side of the device to fully saturate the magneto-optical film with magnetization. With horizontal magnetization, vertically asymmetrical MO waveguides can generate NRPS. However, achieving NRPS in the TE mode necessitates vertical magnetization and horizontal asymmetry, which can pose challenges in manufacturing. Integrating a TE-TM converter before the TM mode isolator offers an alternative approach to realizing the functionality of the TE mode isolator [38]. The operating principle is illustrated in Figure 1b,c. During forward light transmission, by optimizing the length of the upper and lower waveguide arms, the RPS waveguides can generate a phase difference of π/2 + 2 mπ, while the NRPS waveguides create a phase difference of −π/2. The phase difference is added to 2 mπ. Light undergoes constructive interference within the right MMI coupler, resulting in high transmission. When light is transmitted backward, the RPS waveguides induce a phase difference of π/2 + 2 mπ, while the NRPS waveguides produce a phase difference of π/2. The cumulative phase difference is π + 2 mπ. The light undergoes destructive interference in the left MMI coupler, leading to low transmission.

The length difference of RPS waveguides and the length of NRPS waveguides are determined by the propagation constant:(3)$$L_{NRPS}=\frac{\pi }{2(\beta_{f}-\beta_{b})}=\frac{\pi }{2NRPS}$$
(4)$$\Delta L_{RPS}=\frac{(4m+1)\pi }{2\beta_{0}}$$
where $\beta_{0}$ is the propagation constant of the InP waveguide. ***m*** is an integer, and different values of ***m*** correspond to different $\Delta L_{RPS}$. The performance of the isolator, including isolation and bandwidth, is largely affected by the selection of waveguide arm length. Achieving precise waveguide arm dimensions during actual manufacturing is crucial for ensuring high-quality transmission performance. Hence, the transmission performance of a device can be notably influenced by various manufacturing processes, which we will discuss in detail in the following sections.

## 3. Results and Discussion

### 3.1. NRPS, Loss, and Waveguide Structural Determination

We designed two waveguide structures on the InP-on-insulator platform, corresponding to the feasible processes of direct bonding [39,40] and BCB adhesive bonding [36,41], respectively, as shown in Figure 2a,b. For the direct-bonded waveguide structure, a single crystalline layer of Ce:YIG was grown on a (111)-oriented substituted gadolinium gallium garnet (SGGG) substrate. The surface of the target wafer was then activated by O_2_ plasma at 100 W in RF power. Subsequently, the activated surface was brought into contact with InP and strengthened their bond under uniform pressure. Detailed procedures are documented in [40]. Given the substantial difference in thickness between the SGGG substrate and the Ce:YIG film, its removal proves challenging. Therefore, SGGG was used as the upper cladding for simulation and discussion. In the BCB-bonded waveguide structure, a layer of SiO_2_ was first deposited on the InP waveguide. Then, a layer of adhesion promoter AP3000 was applied and gently baked at 135 °C for 5 min to enhance the adhesion of BCB. Finally, under vacuum and continuous pressure, the Ce: YIG mold was placed into contact with the upper surface of the BCB [36]. Both waveguide structures exhibited asymmetric refractive index distributions in the y-direction and the underlying SiO_2_ cladding was thick enough to block out the influence of the substrate.

At a wavelength of 1550 nm, the refractive indices of InP and SiO_2_ were $n_{INP}=3.17$ and $n_{SiO_{2}}=1.45$, respectively. Ce:YIG had a Faraday rotation coefficient of $\theta_{F}=-4500\;{}^{\circ}/\mathrm{cm}$ and a refractive index of $n_{Ce:YIG}=2.2$. The variation in NRPS with different InP thicknesses and widths of the two waveguides was simulated, as illustrated in Figure 2c,d. We used the finite element method (FEM) with ideal conductor boundary conditions in COMSOL and partitioned the two-dimensional waveguide structure into 76,772 elements and 1364 boundary elements. The direct-bonded waveguide exhibited a maximum NRPS of 6488.34 rad/m, whereas the BCB-bonded waveguide showed only 2136.94 rad/m, nearly three times lower. The impact of varying SiO_2_ and BCB thicknesses on NRPS was evaluated, as depicted in Figure 3. When other factors were fixed, NRPS increased as the thickness of the adhesive decreased. Equation (2) implies that the NRPS of the TM mode depends on the gradient distribution of the H_x_ components along the y-direction across the waveguide cross-section. The H_x_ mode field distributions of direct-bonded waveguides and BCB-bonded waveguides with $H_{SiO_{2}}=30\;\mathrm{nm}$, $H_{BCB}=50\;\mathrm{nm}$ and $H_{SiO_{2}}=70\;\mathrm{nm}$, $H_{BCB}=100\;\mathrm{nm}$ are depicted in (1), (2), and (3) in Figure 3. The refractive indices of SiO_2_ and BCB are smaller than those of Ce:YIG and InP, which leads to a decrease in the evanescent field entering the Ce:YIG layer. The upper and lower interfaces of the Ce:YIG cladding exhibit a smaller H_x_ gradient, resulting in a reduced NRPS. Direct-bonded waveguides can be conceptualized as a scenario where the thickness of the adhesive used is zero, thus yielding a higher NRPS.

The propagation loss of the TM mode light in the NRPS waveguide is defined as follows:(5)$$\alpha_{NRPS}=\Gamma_{InP}\times \alpha_{InP}+\Gamma_{Ce:YIG}\times \alpha_{Ce:YIG}+\Gamma_{Clad}\times \alpha_{Clad}$$
where $\alpha_{InP}$, $\alpha_{Ce:YIG}$, and $\alpha_{clad}$ represent the optical absorption loss of the InP waveguide, Ce:YIG film, and cladding materials, respectively. The loss of the InP waveguide based on the Si substrate is assumed to be 4 dB/cm [42], while Ce:YIG has a loss of 42 dB/cm [43]. The propagation losses of SiO_2_, SGGG, and DVS-BCB are small enough to be regarded as 0 at the 1550 nm wavelength. $\Gamma_{InP}$, $\Gamma_{Ce:YIG}$ and $\Gamma_{Clad}$ denote the confinement factors of the composite waveguide, which can be calculated as follows:(6)Γ=nc0ε0∬E2dxdy∬∞Re(E×H*)⋅z˜⋅dxdy

We simulated the Γ of each material in the two composite waveguides, as listed in Table 1. The two structures had MO waveguides of lengths 242.442 µm and 735.068 µm, resulting in losses of 0.687 dB and 1.064 dB. Considering the compactness and loss of the device, the direct-bonded waveguide structure in Figure 2a was adopted for subsequent calculations.

Figure 4 illustrates the variation in NRPS and H_x_ mode field distribution with the thickness of Ce:YIG. As the Ce:YIG thickness increases, NRPS shows a trend of first increasing and then flattening. When analyzing the magnetic field distribution, it is evident that the majority of the H_x_ field gradually becomes contained within the Ce:YIG layer until it is completely contained. The integrated area of the H_x_ magnetic field in the Ce:YIG layer tends to be constant. An MO layer thickness of 500 nm was selected. At this thickness, a large NRPS could be obtained, while also being sufficient to protect the evanescent field from the SGGG substrate interference. Additionally, the flat top of the NRPS curve indicates good manufacturing tolerances.

The RPS waveguide with a ridged waveguide structure is composed of an InP core and a SiO_2_ substrate. The structure and the mode field distribution of the RPS waveguide are shown in Figure 5a,b. It should be noted that the size of the InP must match that of the NRPS waveguide ($W_{InP}=450\;nm$, $H_{InP}=240\;nm$) to minimize mode mismatch at the junction. Due to different transmission modes in RPS and NRPS waveguides, there is modal overlap at the joint, leading to coupling loss. We simulated the mode field distribution at the two waveguide interfaces for the RPS and NRPS waveguides, considering both identical and different InP core sizes. Coupling loss is pronounced when there is a discrepancy in size.

### 3.2. Isolation, Bandwidth, and Free Spectral Range

According to Equation (3), when NRPS is 6488.34 rad/m, the NRPS waveguide length can be calculated as L_NRPS_ = 242.4418 µm. Based on the adjustable length difference between the upper and lower RPS waveguides, length differences of ***m*** = 0, ***m*** = 5, and ***m*** = 15 in Equation (4) are calculated as $\Delta L_{RPS(m=0)}=0.2561\;\mu m$, $\Delta L_{RPS(m=5)}=5.3786\;\mu m$, and $\Delta L_{RPS(m=15)}=15.5624\;\mu m$. For different ***m*** values, the transmittance of the device at a wavelength from 1510 nm to 1590 nm was calculated according to Equation (5), as shown in Figure 6a. The red curve represents forward transmission, while the blue curve represents backward transmission transmittance.
(7)$$T_{FOR,BACK}=10\log_{10}\frac{1+\cos(\mp NRPS(\lambda )\times L_{NRPS}+\beta_{0}(\lambda )\times \Delta L_{RPS})}{2}$$

A large RPS waveguide length difference makes the device more dependent on the wavelength. At $\Delta L_{RPS}=0.26\;\mu m$, the 30 dB bandwidth exceeds 70 nm. With an increase in $\Delta L_{RPS}$ to 15.56 µm, the 30 dB bandwidth narrows to just 1.5 nm. Additionally, increased $\Delta L_{RPS}$ leads to reduced isolation. The difference in length of the waveguide arm affects the free spectral range (FSR) of the MZI transmission spectrum. A broader FSR corresponds to higher isolation across a wider wavelength range, indicating greater bandwidth. Figure 6b shows the FSR variation in the device with different ***m*** values. Smaller ***m*** values correlate with larger FSRs and better isolation performance. Moreover, when the value of ***m*** is less than 40, the bandwidth becomes more sensitive to the difference in waveguide arm length.

### 3.3. Manufacturing Accuracy and Error

The size and propagation constant of the waveguide arms play a crucial role in phase matching. Hence, precise control over waveguide size during manufacturing is essential to achieve optimal isolation and bandwidth. We simulated isolator reverse transmission curves with varying waveguide arm manufacturing accuracies, as shown in Figure 7. It is evident that low precision leads to a shift in the center wavelength of the reverse isolation, with this shift diminishing as the length difference of the RPS waveguide arms increases. This situation can be analyzed according to Equation (5). If the values of $L_{NRPS}$ and $\Delta L_{RPS}$ cannot be determined precisely, the desired phase difference at 1550 nm wavelength cannot be achieved by multiplying waveguide arm length by NRPS and $\beta_{0}$, but it can be achieved at other wavelengths. This results in the maximum isolation of the curve occurring at other wavelengths, causing drift.

The precision of the waveguide arm length primarily impacts the maximum isolation of the device and does not directly affect its bandwidth. By enhancing the manufacturing accuracy, the isolation can be improved, provided that the device transmission curve does not shift. This improvement is attributed to the more accurate waveguide arm achieving a phase difference closer to the theoretical value. At ***m*** = 15, the waveguide arm’s accuracy causes a difference in isolation of approximately 40 dB.

Variations in propagation constants can also lead to shifts in the wavelength of the isolator away from the center of backward loss. This is mainly reflected in the error with the thickness and width of the InP layer. Figure 8 illustrates the deviation of the reverse transmittance as a function of the errors in InP width and thickness. It is found that the center wavelength shift is directly proportional to InP core size errors, and is more sensitive to thickness errors. This sensitivity arises because changes in thickness can significantly impact the H_x_ field entering the Ce:YIG layer, thereby influencing the propagation constant and NRPS.

### 3.4. Manufacturing Processes and Tolerances

Manufacturing accuracy significantly impacts phase deviations in isolators, making the selection of an appropriate manufacturing process critical for achieving optimal isolation performance. Currently, widely used processes include photolithography, nanoimprint lithography (NIL), and electron beam lithography (EBL). Manufacturing films smaller than sub-micron in size through photolithography is challenging. The accuracy of NIL is considerable, which can reach up to tens of nanometers. EBL represents the smallest and thinnest utility pencil known to be capable of producing pattern features down to a few nanometers in size. Furthermore, improvements have been made to address the challenge of heightened losses caused by lens aberrations and motorized stage instability [44]. Taking ***m*** = 5 as an example, Figure 9 illustrates the impact of the maximum error in waveguide arm length on the simulated transmission center wavelength shift for various processes, focusing solely on reverse transmittance. With a more precise manufacturing process, the three curves coincide more closely, resulting in reduced transmittance shifts due to errors. For EBL, the maximum deviation of the curve is only 1 nm, indicating a negligible curve shift caused by maximum process error. However, high-precision processes are expensive, inefficient, and unsuitable for commercial applications. Therefore, during the actual manufacturing process, a comprehensive assessment is necessary to select a suitable process.

Figure 10 illustrates the manufacturing tolerance required to achieve an isolation of ≥20 dB and ≥25 dB at a wavelength of 1550 nm using different processes. When employing photolithography technology, the required degree of isolation cannot be achieved by the theoretical value of the RPS waveguide length difference. However, this shortfall can be compensated for by adjusting the actual length of the NRPS waveguides. In Figure 10a,b,e, we calculate this compensation, providing a reference for actual manufacturing. The solid lines represent achievable isolation requirements. Higher isolation requirements correspond to smaller manufacturing tolerances. Isolators with large waveguide arm length differences are more adaptable to low-precision processes. In summary, if high accuracy is achievable in manufacturing, designs with small waveguide arm differences are advantageous. However, when process accuracy is limited, we need to compromise on the bandwidth and choose a larger RPS waveguide arm difference.

## 4. Conclusions

In summary, we propose an MZI-type basic TM mode magneto-optical isolator leveraging InP/SiO_2_ heterogeneous integration technology. Using two achievable processes of integrating InP and MO materials, two NRPS waveguide structures were constructed. The direct-bonded waveguide exhibits three times more NRPS than the BCB-bonded waveguide. Furthermore, given its smaller footprint and lower loss, the direct-bonded waveguide was ultimately chosen. Through waveguide optimization, we obtained a maximum NRPS of 6488.34 rad/m when $W_{InP}=450\;nm$, $H_{InP}=240\;nm$, and $H_{Ce:YIG}=500\;nm$. At a wavelength of 1550 nm, the device has a maximum 30 dB bandwidth of 72 nm. The reported NRPS and bandwidth of this work are superior to those of isolators based on other platforms [8,23,26,45]. The effects of length difference of the waveguide arm and dimension errors of the waveguide core layer on the isolation bandwidth and the central wavelength of transmittance were analyzed. Manufacturing tolerances for three representative accuracies of processes were calculated based on these simulation results. With its substantial operating bandwidth, high isolation, compact footprint, and significant manufacturing tolerance, this optical isolator holds promise for applications in the realm of non-reciprocal photonic devices. Moreover, its integration into the InP-on-insulator platform offers an innovative approach to achieving the integration of both active and passive components.

## Figures and Tables

**Figure 1 nanomaterials-14-00709-f001:**
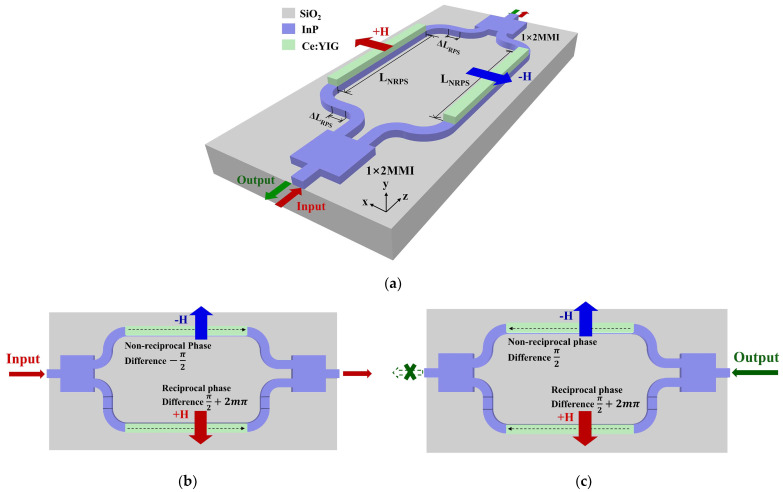
Overall sketch and operating principle of MZI magneto-optical isolator. (**a**) Overall sketch; (**b**) forward transmission; (**c**) backward transmission.

**Figure 2 nanomaterials-14-00709-f002:**
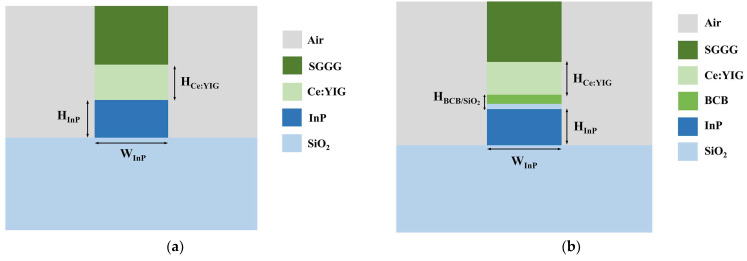
(**a**,**b**) Cross-sections of the direct-bonded and BCB-bonded waveguides. (**c**,**d**) NRPSs of direct-bonded and BCB-bonded waveguides with different InP thicknesses and widths.

**Figure 3 nanomaterials-14-00709-f003:**
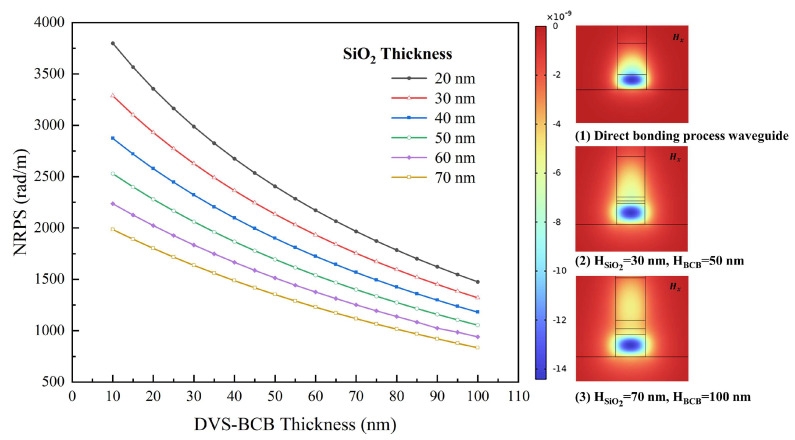
NRPS and H_x_ mode field distributions for different adhesive thicknesses.

**Figure 4 nanomaterials-14-00709-f004:**
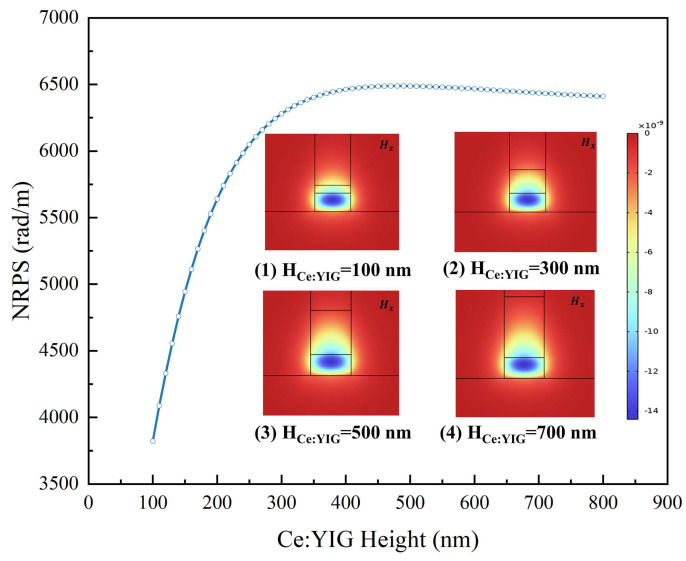
The field distribution of the NRPS and H_x_ modes of the direct-bonded waveguide as a function of the thickness of Ce:YIG.

**Figure 5 nanomaterials-14-00709-f005:**
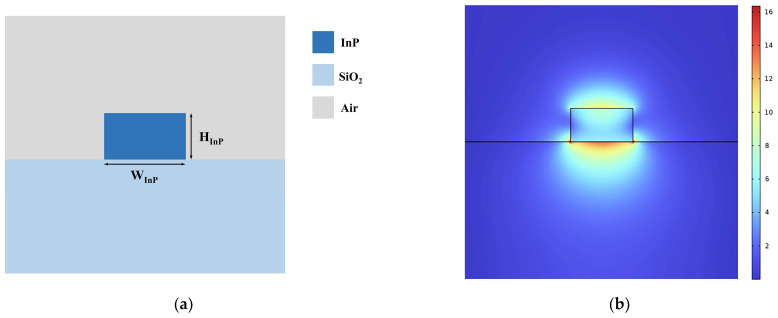
(**a**) Cross-section of the RPS waveguide. (**b**) Transverse electric field mode field distribution of the RPS waveguide. (**c**) Magnetic field distribution at the interface when the InP cores of two waveguides are of equal size. (**d**) Magnetic field distribution at the interface when the InP cores of two waveguides are of different sizes.

**Figure 6 nanomaterials-14-00709-f006:**
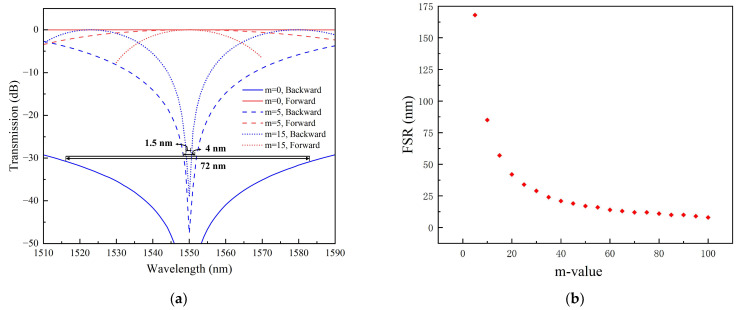
(**a**) Transmission spectra of the isolator with different ***m*** values in the forward and backward propagating directions. (**b**) FSR varies with different ***m*** values.

**Figure 7 nanomaterials-14-00709-f007:**
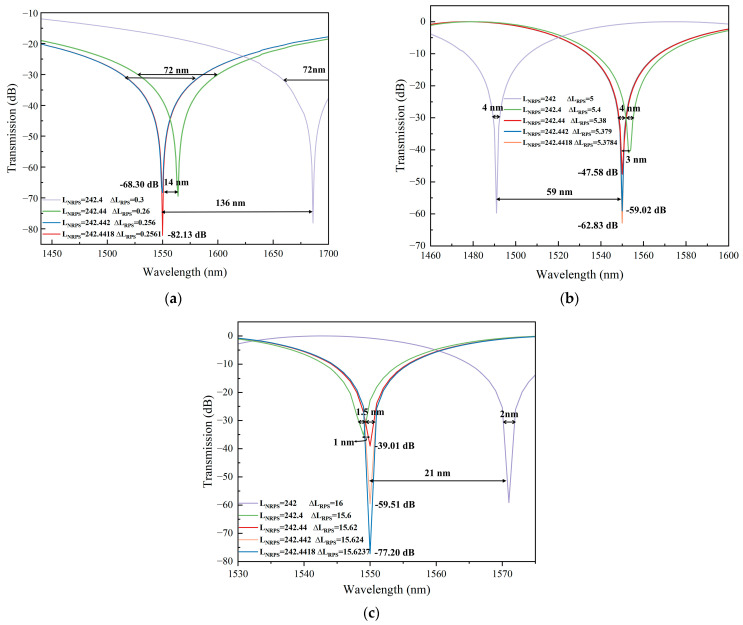
Transmission spectra of isolators with different manufacturing accuracies. (**a**) ***m*** = 0; (**b**) ***m*** = 5; (**c**) ***m*** = 15.

**Figure 8 nanomaterials-14-00709-f008:**
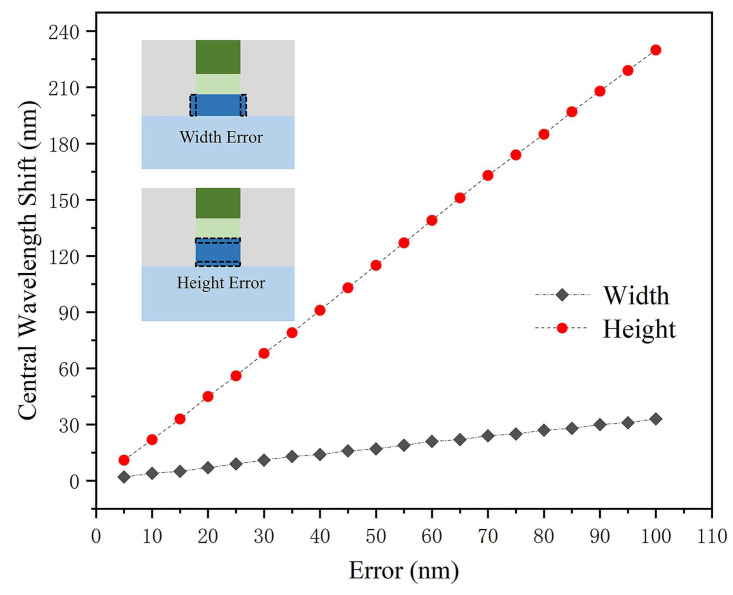
Shift in the transmittance center wavelength caused by errors in the length and width of the InP core.

**Figure 9 nanomaterials-14-00709-f009:**
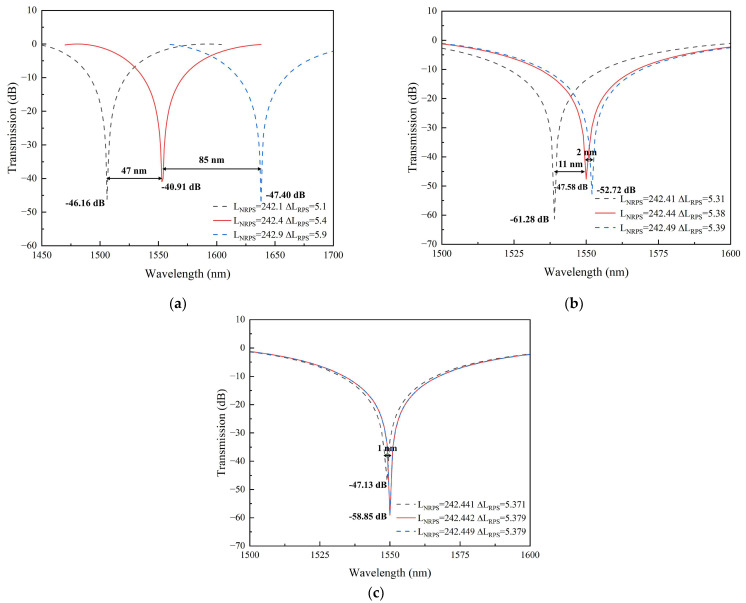
The transmission spectral shift caused by the maximum errors of different processes. (**a**) Photolithography; (**b**) NIL; (**c**) EBL.

**Figure 10 nanomaterials-14-00709-f010:**
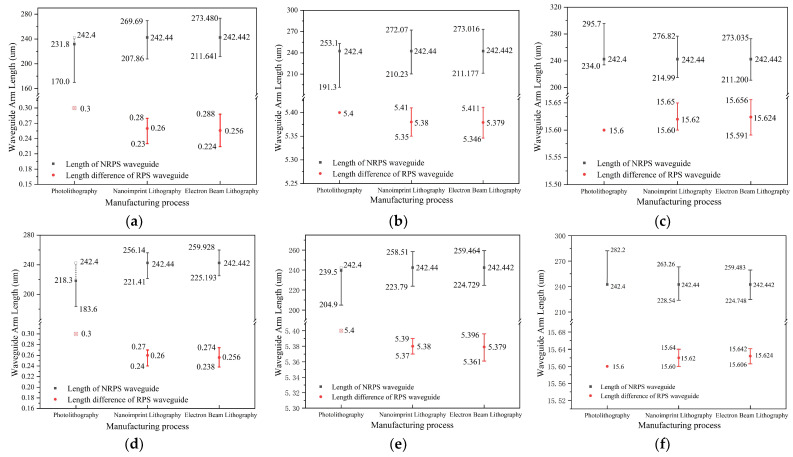
Tolerance of waveguide arms with different processes when the isolation at 1550 nm wavelength ≥ 20 dB. (**a**) ***m*** = 0; (**b**) ***m*** = 5; (**c**) ***m*** = 15. Tolerance of waveguide arms with different processes when the isolation at 1550 nm wavelength ≥ 25 dB. (**d**) ***m*** = 0; (**e**) ***m*** = 5; (**f**) ***m*** = 15.

**Table 1 nanomaterials-14-00709-t001:** The confinement factors of two composite waveguides.

Material	Propagation Loss (*α*)	Direct-Bonded Waveguide	BCB-Bonded Waveguide
		Confinement Factors (Γ)	
InP	4 dB/cm	45.5754%	39.5930%
Ce:YIG	42 dB/cm	63.1869%	30.6780%
SiO_2_	~0 dB/cm	15.4238%	35.9880%
SGGG	~0 dB/cm	1.6389%	3.3344%
BCB	~0 dB/cm	/	16.5910%

## Data Availability

Data are contained within the article.

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
