# Peer review of "On-Chip Broadband, Compact TM Mode Mach–Zehnder Optical Isolator Based on InP-on-Insulator Platforms"

_nanomaterials, 2024, doi:10.3390/nano14080709_

Round 1
Reviewer 1 Report
Comments and Suggestions for Authors
The paper investigates a concept for an optical isolator that has already been shown on silicon photonics platforms, but transports it to an InP-based platform. Also there some earlier work is done. So the content can not be described as original, but does have some value as exploration.
I noticed the following points, which should be addressed and improved by the authors to warrant publication.
1) The motivation is the monolithic integration of an isolator with lasers. However, the authors neglect that lasers usually operate in the TE-mode, so that a TM-mode isolator does not offer any protection. Actually, one would like to isolate all backward light from the laser. How can this device help there?
2) In the analysis the losses are neglected. The MO-material used is not completely transparant, but does result in some propagation loss. It should be taken into account, since it could also influence the conclusions (e.g. regarding the two bonding variation discussed).
3) Also, the results obtained are unrealistic for practical devices. The authors do discuss fabrication tolerances, but neglect with their high isolation figures that real-world MZIs do not reach extinction beyond say 25 dB. The reason is that the two arms are always different in transmission loss ( and phase errors). Also, the MMI couplers used have a certain level of reflection, which directly influences the isolation function. These issues should be discussed.
4) The magnetization is also not discussed. How do the authors magnetize two closely placed pieces of Ce:YIG? And in terms of fabrication: how do you place them accurately on the waveguides?
5) the carrier material of the CE:YIG layer seems to be neglected in the simulations. It has different optical properties, and therefore will influence the mode fields and the propagation losses.
6) The authors state ( top of p.7), that the RPS-waveguide must have the same dimensions as the NRPS-waveguide, to avoid mode mismatch. However, the modes are quite different in these waveguides (due to the presence of the MO-cladding). Thus a severe coupling loss (and reflection) can be expected. I suggest to simulate this and try to optimize the coupling.
7) In the tolerance analysis the motivation is missing for the specifications. E.g., with a 30 dB bandwidth of 72 nm a requirement of <3 nm wavelength shift seems overdone.
Comments on the Quality of English LanguageThe English is understandable, but still contains many errors. I suggest that the authors have their text checked by a person skilled in the language.
Reviewer 2 Report
Comments and Suggestions for Authors
The paper is dedicated to design of broadband, compact TM mode Mach–Zehnder optical isolator based on InP-on-Insulator platforms. Authors compared non-reciprocal phase shift of the direct-bonded waveguide and the DVS-Benzocyclobuten-bonded waveguide. The direct-bonded waveguide structure with a length difference of 0.256 um achieved a 30 dB bandwidth of 72 nm at a wavelength of 1550 nm.
Authors present important research for optical communication systems. Although the data should be presented clearer. The following could help authors:
1. Abstract is too messy. Better logic should be provided. Author state on comparation of materials, but present results only for one.
2. Line 49 “In this work, we compare two bonding methods”. I suppose that authors compare designs, not methods.
3. “Device Structure and Principle” section needs formulated task and result
4. Figure 1 – fonts should be bigger.
5. All subsections in section 3 should have clear results and comparation of 2 materials
6. In conclusion authors do not compare their results with already published.
7. For EBL fabrication of photonic structure authors could check the following article https://doi.org/10.1364/OE.477458
Comments on the Quality of English Language1. Line 49 “In this work, we compare two bonding methods”. I suppose that authors compare designs, not methods.
All text should be checked for such wording misunderstanding.
Reviewer 3 Report
Comments and Suggestions for Authors
I have the following suggestions:
1) The author should explain in detail which scenarios the isolator can be used in the PICs.
2) The E-field or H-field distribution in the device should be shown from the top view where forward propagation and back isolation should be visible.
3) For different wavelength ranges, let's say 1310 nm, the device has to be optimized again. Or is there any other mechanism where isolator can become wavelength-independent?
4) Why does the isolator only work for TM polarization? Can it be employed for TE polarization? And what is the procedure to fabricate this device for polarization-independent action?
5) The author has not presented the details of the simulation model, boundary conditions, meshing, etc.
6) In Figure 3 and Figure 4, the color bar must be presented with the E-field distribution of the mode in the waveguide.
Comments on the Quality of English Languagenone.
Round 2
Reviewer 1 Report
Comments and Suggestions for Authors
Thanks to the authors for improving their paper. The remaining issues are minor and mostly covered, according to the reply, by future work. I therefore can accept publication.
Reviewer 2 Report
Comments and Suggestions for Authors
All my comments were taken into account. Revised article could be published
Comments on the Quality of English LanguageRevised article could be published
Reviewer 3 Report
Comments and Suggestions for Authors
I am willing to accept the paper in its current form.
Comments on the Quality of English LanguageNone.